# Organic Nitrogen Supplementation Increases Vegetative and Reproductive Biomass in a Versatile White Rot Fungus

**DOI:** 10.3390/jof9010007

**Published:** 2022-12-20

**Authors:** Florian Hennicke, Lena Fleckenstein, Claus Bässler, Franz-Sebastian Krah

**Affiliations:** 1Project Group Genetics and Genomics of Fungi, Chair Evolution of Plants and Fungi, Ruhr-University Bochum (RUB), 44801 Bochum, Germany; 2Conservation Biology, Institute for Ecology, Evolution and Diversity, Faculty of Biological Sciences, Goethe University Frankfurt, 60323 Frankfurt am Main, Germany; 3Bavarian Forest National Park, 94481 Grafenau, Germany

**Keywords:** Basidiomycota, Agaricomycetes, Pioppino, model system, experiment, resource, adenosine

## Abstract

The Black Poplar Mushroom *Cyclocybe aegerita* (syn. *Agrocybe aegerita*) is a white-rot fungus that naturally fruits from woody substrates, including buried wood. It is known for its substrate versatility and is equipped with a respective carbohydrate-active enzyme repertoire being intermediate between typical white-rot fungi and plant litter decomposers. Given relative nitrogen scarcity in wood, mobilization of nitrogen from surrounding litter is known as a way to meet nitrogen requirements for cellular homeostasis and reproduction of wood decay fungi. However, the effect of added nitrogen on vegetative and reproductive biomass has not yet been studied in a uniform minimalistic laboratory setup. For *C. aegerita*, such a growth and fruiting setup has been developed. In the present study, this white-rot fungus has been grown with and without additional β-adenosine, an organic nitrogen source present in plant litter. Elevated β-adenosine levels increased aerial mycelium weight by 30% (1 × β-adenosine) and 55% (10 × β-adenosine), reproductive biomass by 75% (1 × β-adenosine) and by 100% (10 × β-adenosine), number of primordia by 127% (10 × β-adenosine) and accelerated primordium formation by 1.6 days (10 × β-adenosine), compared to the control treatment. These findings imply that *C. aegerita* invests additional organic nitrogen resources into direct vegetative and reproductive biomass build-up at the same time. Colonization of niches with accessory nitrogen sources, like buried wood, which is near the plant litter layer, may thus provide an evolutionary fitness advantage. Globally anthropogenically altered nitrogen dynamics may affect hyphal-driven processes as well as fruit body-driven food webs.

## 1. Introduction

On Earth, terrestrial biomass is mainly composed of lignocellulose from plant material that chiefly consists of lignin, cellulose, and hemicellulose [1]. Carbon from this polymer is mineralized by wood- and litter-decomposing mushroom-forming fungi of the Basidiomycota class Agaricomycetes [2]. Other than brown-rot fungi or litter decomposers, white-rot fungi like the genome-sequenced [3] cultivated edible agaric *Cyclocybe aegerita* (V. Brig.) Vizzini (syn. *Agrocybe aegerita* (V. Brig.) Singer) can effectively degrade lignin. This breakdown depends on the co-metabolism of cellulose as an energy and carbon source, with nitrogen playing a regulatory role [2,4]. In contrast to soil, which provides nitrogen mainly as organic compounds, wood is relatively poor in nitrogen [5,6]. However, sufficient nitrogen acquisition is required for wood decay by nutrient-costly extracellular enzyme production [7], for cellular homeostasis, and fungal reproduction. Therefore, ways to circumvent nitrogen scarcity in wood have evolved wood-decaying species like *C. aegerita,* which also fruit from buried wood [8]. This is achieved, for example, by mobilization of accessory organic but also inorganic nitrogen sources from surrounding soil and (nitrogen-rich) plant litter, or by interaction with atmospheric N_2_-assimilating (diazotrophic) bacteria [6,9,10,11,12,13,14,15]. Still, it has not yet been comprehensively explored in a uniform minimalistic setup allowing traversal of the whole fungal life cycle, to what degree accessory nitrogen is converted into vegetative and/or reproductive biomass increase (mushroom formation).

Sexual reproduction (fruiting) of mushroom (fruiting body)-forming fungi is triggered in a species-specific manner by diverse abiotic and biotic factors, including nutrients or hormone-like compounds [16,17,18,19]. Hence, growth and fruiting of mushrooms are greatly affected by the composition of the substrate, as the main source of nutrients [20,21,22]. The effect of substrate supplementation on growth or fruiting has been monitored in a couple of individual ecologically or economically relevant edible mushroom species [10,23,24]. Despite the described insights, previous studies also contain strong limitations, such as the medium requirements for getting the respective fungi to fruit, which involves complex wheat straw-, sawdust- or cereal-based substrates. To overcome these limitations, in the present study, a uniform, axenic, minimalistic laboratory growth- and fruiting setup is applied using a well-established model system for mushroom formation, *Cyclocybe aegerita* [25,26,27,28]. Using agar medium supplementation with a source of accessory organic nitrogen (in the form of the nucleoside β-adenosine), allowed us to assess how far this fungus profits from additional nitrogen mobilization. To quantify profitability, different fungal growth parameters were measured: mycelium growth, aerial mycelium biomass, the number, and timing of primordia formation, as well as reproductive fruit body biomass.

## 2. Material and Methods

### 2.1. Strains, Culture Maintenance, Growth Rate and Fruiting Setup

The genome-sequenced dikaryon *Cyclocybe aegerita* AAE-3 [3,27] was used for all experiments. Routine propagation and long-term storage were carried out as described previously [27]. For fruiting, replicate 1.5% MEA plates were employed (without accessory organic nitrogen supplement or with 15 µg/mL β-adenosine [‘adenosine 1×’, moderate supplementation] and, respectively, 150 µg/mL β-adenosine [‘adenosine 10×’, more intense supplementation]; β-adenosine was purchased from Carl Roth GmbH & Co. KG, Karlsruhe, Germany, cat. No. A9251-5G). The MEA agar solution was autoclaved and rested until cooled to ca. 40 °C. The β-adenosine solution was filter-sterilized using 10 mL disposable syringes (Cat. #612-2893 VWR International, Radnor, PA, USA) with sterile syringe filters of 0.2 µm cellulose acetate (Cat. #514-0073 VWR International, Radnor, PA, USA), and added to the agar solution after the agar cooled to ca. 40 °C. These plates were centrally inoculated with a mycelium-overgrown agar plug (0.3 cm diameter, punched out from the edge of a freshly colonized 1.5% MEA plate). Repeated three times independently, during each fruiting experiment, 14 replicate plates were used to monitor the growth rate of *C. aegerita* AAE-3. All other replicate plates were wrapped in aluminum foil and grown at 25 °C in the dark until the growth control plates reached the plate edge. Fully-colonized plates were then transferred to the fruiting conditions specified in Ref. [29]. Fruiting experiments were repeated three times independently for each treatment, while each round comprised 81 replicate plates for each treatment. An illustration of the experimental setup is included in the Appendix A.

### 2.2. Assessment of Mycelial Growth and Reproductive Success

To gain estimates of mycelial growth, the mycelium growth rate as well as aerial mycelium dry weight was measured. The growth rate was assessed based on daily increment measures until the mycelium reached the edge of the agar plate. In detail, we marked the additional day-to-day growth on the bottom side of the respective replicate plates. We determined the aerial mycelium dry weight by carefully scraping off the aerial mycelium at the time point when the mycelium reached the edge of the plate. Aerial mycelium was then transferred into a 2.0 mL Eppendorf tube and dried for 24 h in a drying oven at 40 °C. Then dry weight was determined via a weighing balance (LECO Corporation 150, mod. nr. 751–200, serial nr. 284, four digital points) by measuring the dried mycelium without an Eppendorf tube.

To monitor reproductive success, the following indicators were measured: the number of primordia produced, the time from when fruiting had been induced until the first primordia emerged, and the dry weight of mature fruiting bodies. The number of all newly formed primordia was counted during a period of 40 days after fruiting induction recording the timepoint when the first primordia developed in days after fruiting induction. Furthermore, for fruit bodies that reached maturity (spore release was visible by dark deposits in the plates), dry weight was determined. We cut the mature fruit bodies at the very base of the stipe and transferred them to empty Petri dishes, and dried them for 24 h in a drying oven at 40 °C. Then, dry weight was determined via a weighing balance (as above) by measuring the dried fruiting body mass.

### 2.3. Statistical Analysis

To test for significant differences in biomass means between the treatment factors, linear mixed-effects models implemented in the add-on R package lme4 were employed [30]. In total, five models with different response variables were fit using the same predictor variable and random effect. As the response variables, the daily growth rate (mm), the aerial mycelium dry weight per plate (g), the days since fruiting induction until the first primordium/primordia emerged, the number of primordia per plate, and the reproductive biomass per plate were used. As the predictor variable, we employed the factorial adenosine treatment with three levels (control, adenosine 1 × [15 µg/mL], adenosine 10 × [150 µg/mL]). The chamber was used as a random factor because the experiment was replicated in three climate chambers. Post-hoc to this model, Tukey tests were performed to test the significance between all pairwise treatment levels using the R add-on package multcomp [31].

## 3. Results

No significant difference in daily growth rate was found between treatments (Figure 1A). Nonetheless, a significantly increased aerial mycelium dry weight of the adenosine-supplemented cultures was detected compared to the control (Figure 1B). In contrast to the control and the moderately adenosine-supplemented cultures, mycelia that received a more intense adenosine supplementation needed significantly fewer days to develop first primordia after fruiting induction (Figure 1C). On average, the first primordia appeared 8.1 days after fruiting induction on medium with intense adenosine supplementation, compared to 9.7 days on the control plates and 9.2 days on the moderate adenosine supplemented plates. In addition, the number of primordia in cultures with more intense adenosine supplementation was significantly increased compared to control and moderate supplementation (Figure 1D and Appendix A). A significantly higher reproductive biomass per plate was found in cultures with moderate and more intense adenosine supplementation, compared to the control (Figure 1E and Appendix A).

## 4. Discussion

In the present study, it has been found that accessory organic nitrogen in the form of the nucleoside adenosine enhances vegetative (mycelial) and reproductive (mushroom) biomass build-up by a white-rot fungus in a uniform minimalistic setup where it can traverse its entire life cycle.

### 4.1. Effect of Adenosine Addition on Mycelium and Primordia Formation

In previous work [10,32] where β-adenosine or β-adenosine-/nitrogen-rich rye grass material/extract was used as a nitrogen additive, only a partial traversal of the mushroom life cycle on agar medium took place. In agreement with these studies, mycelial biomass increased both when adding 15 µg/mL or 150 µg/mL β-adenosine to the agar medium was detected here (see Figure 1B). Domondon et al. [10] saw increased mycelial growth (colony diameter on day 7 in darkness at 24 °C) on agar medium with 30% ryegrass extract in the litter-decomposer *Stropharia rugosoannulata* Farl. ex Murrill, but not for *Pleurotus pulmonarius* (Fr.) Quél. Similar to the data on the white-rot fungus *P. pulmonarius*, here, no differences were seen in mycelial growth rates between the control and the nitrogen-supplemented treatments, in the white-rot fungus *C. aegerita* (see Figure 1A). Still, Domondon et al. [10] noted that *P. pulmonarius* formed dense mycelial cords (rhizopmorphs) on β-adenosine-supplemented plates. If dry weight was assessed, cord formation by this fungus upon β-adenosine supplementation might eventually reveal an increase of mycelial biomass, as detected in the present case (see Figure 1B) and in the ectomycorrhizal (ECM) fungus *Suillus luteus* [32]. Other than Zhang et al. [32], where 32 µg/mL β-adenosine already led to a dry weight drop over 16 µg/mL, here such an ‘overdosage effect’ was not observed when increasing the amount of β-adenosine supplementation 10×, i.e., to 150 µg/mL (see Figure 1B).

The same applies to the β-adenosine-triggered acceleration of primordia formation on agar medium in *C. aegerita* (see Figure 1C) and *P. pulmonarius* [10]: *P. pulmonarius* primordia appeared five days after the agar medium was supplemented with a dosage of 25 µg/mL, compared to a dosage of 12 µg/mL β-adenosine. In contrast, *C. aegerita* primordia appeared circa two days earlier on medium supplemented with 150 µg/mL β-adenosine, compared to circa 10 days in the control. In plates with 15 µg/mL β-adenosine supplementation, primordia arrived after circa nine days with the difference to the control being not significant. Albeit stimulated by orchestrated fruiting cues, *S. luteus* mycelia did not form any fruiting body development stages, regardless of β-adenosine addition in the agar plate-based experimental setup of [32]. Most likely, this is caused by the absence of an ectomycorrhizal partner tree, the presence of which is usually required to trigger fruiting in ECM fungi, under semi-minimalistic laboratory conditions [33].

Accelerated *C. aegerita* primordia formation on a medium with 150 µg/mL β-adenosine was, furthermore, recorded to be accompanied by a higher total count of primordia (see Figure 1D). This is congruent with data for *P. pulmonarius* by Domondon et al. [10], who recorded a higher total count of primordia on β-adenosine-/nitrogen-rich [14] ryegrass-extract agar medium in comparison with the control.

### 4.2. Effect of Adenosine Addition on Reproductive Biomass in Fruit Bodies

Compared to the control, a biomass increase was recorded in mature *C. aegerita* mushrooms from cultures supplemented with 15 µg/mL and 150 µg/mL β-adenosine (see Figure 1E). For *P. pulmonarius*, Domondon et al. [10] could show that only a concentration of 25 µg/mL but not 12 µg/mL β-adenosine would increase mushroom weight significantly in a non-minimalistic setup using wheat straw-based spawn substrate. There, just as in agar medium, an increased concentration of 25 µg/mL β-adenosine led to delayed primordia formation in *P. pulmonarius*. In contrast, supplementation of the medium with 150 µg/mL β-adenosine accelerated primordium formation in *C. aegerita* (see Figure 1C). For *C. aegerita*, these observations also seem to be in good agreement with the results of Uhart et al. [23]. These authors used an East Asian strain of *C. aegerita sensu lato*, *Cyclocybe* sp. 621/04, which, if sequenced, may cluster into *C. chaxingu* agg. [34]. For this strain, a most positive influence of soybean flour, an additive that is rich in organic nitrogen from proteins [35,36], added to wheat straw-based mushroom spawn substrate was recorded on mushroom yield.

### 4.3. Study Limitations and Future Perspective

The present initial analysis provides a new starting point for future work on growth and fruiting in relation to available nitrogen, but it also has some limitations. (i) Mycelium growth and reproductive fruiting should be assessed using different organic as well as inorganic nitrogen species like different amino acids, ammonia and nitrate. This will reveal whether *C. aegerita* prefers reduced organic nitrogen or inorganic N species like ammonia or urea over nitrate for vegetative biomass build-up, as it has been observed with white-rot fungi like *Trametes versicolor* (L.) Lloyd, *Hypholoma fasciculare* (Huds.) P. Kumm., or *Pleurotus ostreatus* (Jacq.) P. Kumm. [4,12]. In the present study, adenosine was chosen as a nitrogen source since nitrogen from soil and plant litter is mainly available in the form of organic nitrogen species. As such, adenosine is abundant and available in litter substrates [10,37], and, thus, was chosen as a nitrogen source also because of its real-world availability to the fungus *C. aegerita*. (ii) Although a standardized MEA medium was employed here, this is still a complex medium and thus, future studies may rather use a minimal medium such as the *Schizophyllum commune* Fr. minimal medium (ScMM) [38]. The nitrogen source in ScMM can be easily modified by exchanging asparagine for other organic or inorganic nitrogen sources. Since *C. aegerita* grows well in liquid ScMM [39], this will allow seeing in how far (a) preferred (in)organic N source(s) will lead to superior whole mycelium biomass increase. (iii) To further understand nitrogen acquisition, the use of microcosms may shed light on the capacities of the fungus to transfer nitrogen across its mycelium from the soil to colonized wood as it has been conducted with notorious white-rot fungi [11,13]. (iv) The present experiment comprised a positive resource gradient with nitrogen addition leading to enhanced growth. Future studies should investigate a negative resource gradient (limitation), which will allow insights into biomass allocation under stressful conditions. Trade-offs might exist between vegetative growth and reproduction [40]. (v) There is an ongoing debate about whether white- and brown-rot wood decay fungi show host specificity towards gymno- or angiosperm dead wood and what the underlying evolutionary process is [41]. Many fungi show such specialization [41] and one explanation might be that gymno- or angiosperm wood differs in nitrogen content affecting growth and fruiting. Future studies should, thus, include multiple brown- and white-rot fungi and test the effect of nitrogen addition on growth and fruiting. Additionally, investigating the average fruit body size on gymno- vs. angiosperm wood species might yield first insights.

On a broader perspective, anthropogenic global nitrogen dynamics are spatially variable [42] but may affect global fungal vegetative and reproductive biomass, with potential effects on hyphal-driven ecosystem processes [43,44], and mushroom-driven food webs [45]. Thus, the issue of how additional nitrogen sources or nitrogen pollution of oligotrophic forest ecosystems will affect the growth of different fungi and species composition should be assessed, e.g., by assays with multiple species in one experimental setup.

## 5. Conclusions

In this laboratory experiment, it has been explored whether the fungus *Cyclocybe aegerita* profits from an additional, real-world nitrogen source. As a result, it has been found that the addition of the nucleoside adenosine enhanced vegetative mycelium, as well as reproductive fruit body biomass, and it led to more and faster primordia formation. This positive resource addition experiment, thus, suggests a fitness advantage of wood decay fungi that retain the ability to mobilize accessory nitrogen resources since larger reproductive success will lead to a production of more sexual spores, and thus, higher genetic diversity.

## Figures and Tables

**Figure 1 jof-09-00007-f001:**
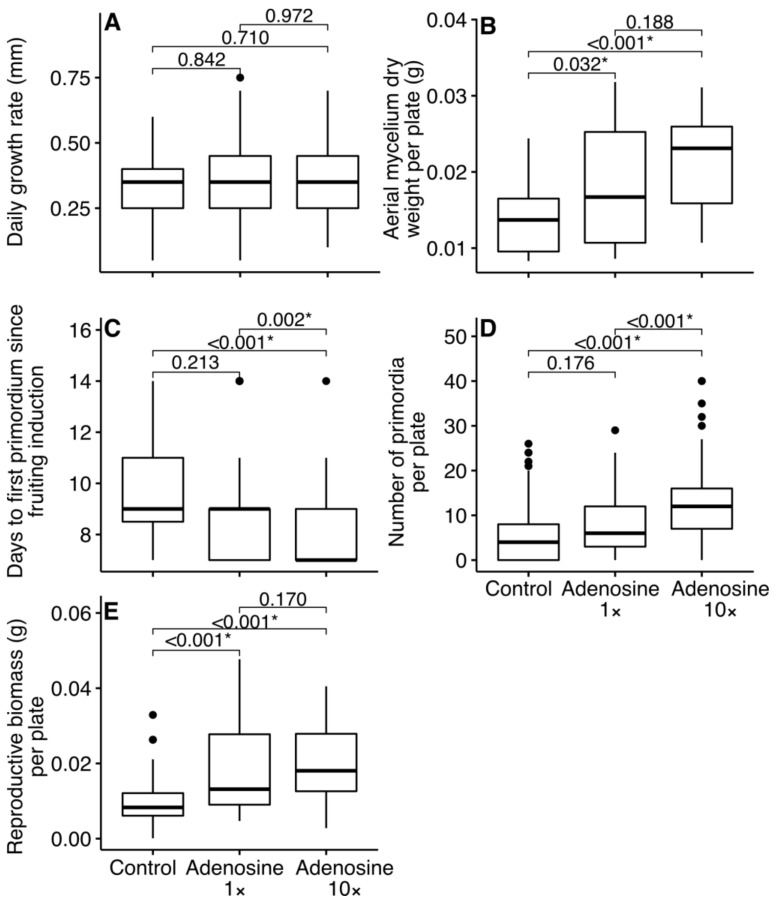
Influence of nitrogen supplementation on fungal vegetative and reproductive biomass of *Cyclocybe aegerita* AAE-3 (**A**) Mycelial growth rate in mm was measured every day (3 experiments with each 14 replicates). (**B**) Aerial mycelium was harvested when mycelium reached the Petri dish edge and then dry-weighed. (**C**) Days until the first primordium was visible after fruiting inoculation; Petri dishes were checked every second day. (**D**) Total number of primordia produced within 40 days after fruiting induction. (**E**) Dry weight of mature fruit bodies (reproductive biomass). (**C**–**E**) Three climate chambers and 81 Petri dishes per treatment were employed to acquire the data. To test for significant differences in means between the treatment factors, linear mixed-effects models with random effect on climate chamber were applied. Asterisks indicate significance at alpha level of 0.05.

## Data Availability

All data generated or analyzed in this study are included in this article and its Appendix A files.

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
