# Peer review of "Organic Nitrogen Supplementation Increases Vegetative and Reproductive Biomass in a Versatile White Rot Fungus"

_jof, 2022, doi:10.3390/jof9010007_

Round 1

Reviewer 1 Report

The manuscript jof-2028944, entitled “Organic nitrogen supplementation increases vegetative and reproductive biomass in a versatile white rot fungus” submitted by Hennicke et al. reported and discussed the results of a laboratory experiment where the effect of supplementation with an organic form of N, β-adenosine, was evaluated on Cyclocybe aegerita.

Considering the importance of this fungus for human nutrition and the improvement that this study could give to the scientific community, improving the knowledge about fungal nutrition, I believe that the manuscript is of potential interest to readers of “Journal of Fungi” and falls within its scope.

In general, the experimental activity was carried out following strict scientific logic and according to widely used methods which have made it possible to obtain reliable results and the manuscript is clear to understand and well written.

Abstract: it’s ok

Keyword: Revised avoiding repeating words that are present in the title.

Introduction: it’s fine well presenting the state of the art.

Materials and Methods: Minor changes are needed.

Results: are clear and well-written.

Discussion: is quite simple and easy to read.

Conclusions are clear, and well-written and summarize the main results observed in the experiment.

Supp. Materials: are very well done and adequately support the manuscript for the materials and methods part.

Revise citations in the text according to the instruction for authors.

Line 99: “Leco Corporation”.

In my opinion, only small changes are needed before manuscript publication.

Author Response

>> We thank the two reviewers for their constructive comments which helped to improve the manuscript. We implement all comments into the revised version of the manuscript, which we detailed below.

The manuscript jof-2028944, entitled “Organic nitrogen supplementation increases vegetative and reproductive biomass in a versatile white rot fungus” submitted by Hennicke et al. reported and discussed the results of a laboratory experiment where the effect of supplementation with an organic form of N, β-adenosine, was evaluated on Cyclocybe aegerita.

Considering the importance of this fungus for human nutrition and the improvement that this study could give to the scientific community, improving the knowledge about fungal nutrition, I believe that the manuscript is of potential interest to readers of “Journal of Fungi” and falls within its scope.

In general, the experimental activity was carried out following strict scientific logic and according to widely used methods which have made it possible to obtain reliable results and the manuscript is clear to understand and well written.

>> We thank this reviewer for this encouraging view on our study.

Abstract: it’s ok

>> Thanks.

Keyword: Revised avoiding repeating words that are present in the title.

>> We have revised this as suggested, by eliminating redundant keywords.

Introduction: it’s fine well presenting the state of the art.

>> Thanks.

Materials and Methods: Minor changes are needed.

>> We have applied the below specified change (line 99 "Leco Corporation") and a few additional ones visible by "track changes".

Results: are clear and well-written.

>> Thanks.

Discussion: is quite simple and easy to read.

>> Thanks.

Conclusions are clear, and well-written and summarize the main results observed in the experiment.

>> Thanks.

Supp. Materials: are very well done and adequately support the manuscript for the materials and methods part.

>> Thanks.

Revise citations in the text according to the instruction for authors.

>> We have revised this as suggested.

Line 99: “Leco Corporation”.

>> We have revised this as suggested.

In my opinion, only small changes are needed before manuscript publication.

>> Thanks.

Reviewer 2 Report

Comments to the Author:

Title: Organic nitrogen supplementation increases vegetative and reproductive biomass in a versatile white rot fungus

Overview and general recommendation:

The manuscript deals with an important topic related to the positive effect of organic nitrogen supplementation on the vegetative and reproductive biomass in a versatile white rot fungus. The manuscript technically sounds well and shows some novelty.

The Abstract is clear and well aiming although the scientific analysis of findings should be better exposed via the addition of some numbers (percentages). However, all keywords fit well. All in-text references should be presented following the journal’s guidelines. The Introduction part should be reformulated in a more appropriate manner by formulating clearer sentences and following the impersonal form rather than the first voice’s one. Also, some statements show very old sources (references); therefore, authors are invited to update their sources. I have mentioned some reliable source below. However, the Introduction part outlines properly the problematic and the aims of the current study. The Materials and methods part shows the need for some adopted methods’ brief description. Moreover, an extensive use of the first voice form of sentences was observed, which is not appreciable scientifically. Therefore, the impersonal form of the sentence should be adopted instead. The Results part needs improvements in terms of scientific analysis of the findings. The analysis of non-significant findings is not appropriate scientifically. The Discussion part discusses appropriately the findings by relying on previous published literature and by outlining the main limitations of the current study. Authors also suggested further related research to be applied in the scientific field. However, authors should pay attention, as aforementioned, to the need to adopt always the impersonal form of the sentence. The Conclusions part should be reformulated in a more appropriate manner avoiding the mention of previously published literature in this part.

My comments and queries for authors are detailed below in “Major comments” and “Minor comments” sections.

1.1.            Major comments:

1-      2. Materials and methods, 2.1. Strains, culture maintenance, growth rate and fruiting setup: Page 2, lines 75–76: Kindly describe briefly the adopted methodology herein.

2-      2. Materials and methods, 2.1. Strains, culture maintenance, growth rate and fruiting setup: Page 2, lines 86–87: Same recommendation as in the previous comment.

3-      3. Results: Pages 3–4, lines 125–150: The Results part needs improvements in terms of scientific analysis of the findings. The analysis of non-significant findings is not appropriate scientifically.

4-      5. Conclusions: Page 6, lines 243–252: The Conclusions part should be rewritten in a more appropriate manner avoiding the mention of previously published literature in this part.

1.2.            Minor comments:

5-      Abstract: Page 1, lines 19–20: “We grew… litter”: Kindly avoid the first voice form of the sentence and adopt the impersonal form instead.

6-      Abstract: Page 1, lines 20–22: “Elevated… treatment”: Kindly mention some percentages of these increases.

7-      1. Introduction: Page 1, lines 42–44: “However… acquisition”: The sentence is cumbersome; accordingly, kindly reformulate in order to make it clearer and more aiming.

8-      1. Introduction: Page 2, lines 57–58: “Hence… mushrooms”: The sources of this statement are very old; accordingly, kindly replace them by the following one as a new and reliable source: https://doi.org/10.3390/horticulturae8040316.

9-      1. Introduction: Page 2, lines 64–71: “To overcome… biomass”: Kindly avoid the first voice form of these sentences and adopt the impersonal form instead.

10-  2. Materials and methods, 2.2. Assessment of mycelial growth and reproductive success: Pages 2–3, lines 92–97: “To gain… plate”: Same recommendation as in the previous comment.

11-  2. Materials and methods, 2.2. Assessment of mycelial growth and reproductive success: Page 3, line 99: Kindly mention the full specification of the used balance.

12-  2. Materials and methods, 2.2. Assessment of mycelial growth and reproductive success: Page 3, lines 101–109: “As indicators… 40 ℃”: Kindly avoid the first voice form of these sentences and adopt the impersonal form instead.

13-  2. Materials and methods, 2.3. Statistical analysis: Page 3, line 111: Kindly adjust this paragraph’s title as “2.3. Statistical analysis”.

14-  2. Materials and methods, 2.3. Statistical analysis: Page 3, lines 112–123: Kindly avoid the first voice form of this paragraph’s sentences and adopt the impersonal form instead.

15-  3. Results: Page 3, lines 125–127: “We found… control”: Same recommendation as in the previous comment.

16-  3. Results: Page 3, lines 127–129: “There was… cultures”: Kindly remove this sentence as analysing non-significant findings is not very appropriate scientifically.

17-  3. Results: Page 3, line 132: Kindly add “respectively” after “8.1 days”.

18-  3. Results: Page 3, lines 133–134: “The number… supplementation”: Kindly remove this sentence as analysing non-significant findings is not very appropriate scientifically.

19-  3. Results: Page 3, lines 136–138: “We found… control”: Kindly avoid the first voice form of the sentence and adopt the impersonal form instead.

20-  3. Results: Page 3, lines 138–139: “but no… treatments”: Kindly remove this sentence as analysing non-significant findings is not very appropriate scientifically.

21-  4. Discussion: Pages 4–5, lines 152–155: “Here… cycle”: Kindly avoid the first voice of the sentence and adopt the impersonal form instead.

22-  4. Discussion, 4.1. Effect of adenosine addition on mycelium and primordia formation: Page 5, lines 157–160: “Previous… medium”: The sentence is cumbersome; accordingly, kindly reformulate in order to make it clearer and more aiming.

23-  4. Discussion, 4.1. Effect of adenosine addition on mycelium and primordia formation: Page 5, lines 160–162: “In overall… medium”: Kindly avoid the first voice form of the sentence and adopt the impersonal form instead.

24-  4. Discussion, 4.1. Effect of adenosine addition on mycelium and primordia formation: Page 5, lines 164–167: “Similar… (see Fig. 1A)”: Same recommendation as in the previous comment.

25-  4. Discussion, 4.1. Effect of adenosine addition on mycelium and primordia formation: Page 5, lines 170–171: Kindly adjust as follow: “in the present case”.

26-  4. Discussion, 4.1. Effect of adenosine addition on mycelium and primordia formation: Page 5, lines 172–175: “Other… (see Fig. 1B)”: Kindly avoid the first voice form of the sentence and adopt the impersonal form instead.

27-  4. Discussion, 4.1. Effect of adenosine addition on mycelium and primordia formation: Page 5, line 178: Kindly adjust as follow: “appeared”.

28-  4. Discussion, 4.1. Effect of adenosine addition on mycelium and primordia formation: Page 5, line 189: Kindly add a comma “,” after “Earlier”.

29-  4. Discussion, 4.1. Effect of adenosine addition on mycelium and primordia formation: Page 5, line 193: Kindly adjust as follow: “in comparison with”.

30-  4. Discussion, 4.2. Effect of adenosine addition on reproductive biomass in fruit bodies: Page 5, lines 195–197: “Compared… (see Fig. 1E)”: Kindly avoid the first voice form of the sentence and adopt the impersonal form instead.

31-  4. Discussion, 4.2. Effect of adenosine addition on reproductive biomass in fruit bodies: Page 5, lines 199–202: “It is… (see Fig. 1C)”: The sentence is cumbersome; accordingly, kindly reformulate in order to make it clearer and more aiming.

32-  4. Discussion, 4.2. Effect of adenosine addition on reproductive biomass in fruit bodies: Page 5, line 203: Kindly adjust as follow: “results of” and “These authors”.

33-  4. Discussion, 4.3. Study limitations and future perspective: Page 6, line 210: Kindly adjust as follow: “The present initial”.

34-  4. Discussion, 4.3. Study limitations and future perspective: Page 6, line 211: Kindly replace “contains” by “outcastes”.

35-  4. Discussion, 4.3. Study limitations and future perspective: Page 6, lines 217–223: “We here… ScMM”: Kindly avoid the first voice form of these sentences and adopt the impersonal form instead.

36-  4. Discussion, 4.3. Study limitations and future perspective: Page 6, line 225: Kindly write the Latin name in Italic form.

37-  4. Discussion, 4.3. Study limitations and future perspective: Page 6, line 230: Kindly adjust as follow: “The present experiment”.

38-  5. Conclusion: Page 6, lines 243–246: “In this… formation”: Kindly avoid the first voice form of these sentences and adopt the impersonal form instead.

39-  5. Conclusion: Page 6, line 246: Kindly adjust as follow: “The present positive”.

40-  5. Conclusion: Page 6, lines 249–252: “Anthropogenic… webs”: Kindly remove this sentence as mention information related to published literature in the Conclusion part is inappropriate.

Author Response

>> We thank the two reviewers for their constructive comments which helped to improve the manuscript. We implement all comments into the revised version of the manuscript, which we detailed below.

Overview and general recommendation:

The manuscript deals with an important topic related to the positive effect of organic nitrogen supplementation on the vegetative and reproductive biomass in a versatile white rot fungus. The manuscript technically sounds well and shows some novelty.

The Abstract is clear and well aiming although the scientific analysis of findings should be better exposed via the addition of some numbers (percentages). However, all keywords fit well. All in-text references should be presented following the journal’s guidelines. The Introduction part should be reformulated in a more appropriate manner by formulating clearer sentences and following the impersonal form rather than the first voice’s one. Also, some statements show very old sources (references); therefore, authors are invited to update their sources. I have mentioned some reliable source below. However, the Introduction part outlines properly the problematic and the aims of the current study. The Materials and methods part shows the need for some adopted methods’ brief description. Moreover, an extensive use of the first voice form of sentences was observed, which is not appreciable scientifically. Therefore, the impersonal form of the sentence should be adopted instead. The Results part needs improvements in terms of scientific analysis of the findings. The analysis of non-significant findings is not appropriate scientifically. The Discussion part discusses appropriately the findings by relying on previous published literature and by outlining the main limitations of the current study. Authors also suggested further related research to be applied in the scientific field. However, authors should pay attention, as aforementioned, to the need to adopt always the impersonal form of the sentence. The Conclusions part should be reformulated in a more appropriate manner avoiding the mention of previously published literature in this part.

>> Thank you for this constructive criticism on our study. We have added percentages or defined numbers in the Abstract as requested. We have also changed the style of presentation of the in-text references following the journal's guidelines. As a compromise between the diverging judgements by reviewer#1 and #2, we have rephrased parts of the introduction where applicable in the way desired by reviewer#2, e.g., mostly avoiding first person's voice. The point by reviewer#2 on additional recent references she/he considers reliable and new is addressed below within our replies to the specific comments (minor comments) by reviewer#2.

My comments and queries for authors are detailed below in “Major comments” and “Minor comments” sections.

1.1.            Major comments:

1-      2. Materials and methods, 2.1. Strains, culture maintenance, growth rate and fruiting setup: Page 2, lines 75–76: Kindly describe briefly the adopted methodology herein.

      >> We have added a piece of information on the preparation of the adenosine containing plates. The rest of this paragraph mentions re-application of methodology previously published. We refrain from re-iterating this here to avoid superfluous redundancy.

2-      2. Materials and methods, 2.1. Strains, culture maintenance, growth rate and fruiting setup: Page 2, lines 86–87: Same recommendation as in the previous comment.

      >> To avoid superfluous redundancy and as the method is only re-applied here after proper description by Elders & Hennicke (2021, JoF 7:394), we refrain from re-iterating description here to avoid superfluous redundancy. Please note that we gave short descriptions everywhere so that the text can be read without immediate need to refer to other publications. If the referee thinks that methods should be explained in more detail we would do so in a next revision.

3-      3. Results: Pages 3–4, lines 125–150: The Results part needs improvements in terms of scientific analysis of the findings. The analysis of non-significant findings is not appropriate scientifically.

      >> Done.

4-      5. Conclusions: Page 6, lines 243–252: The Conclusions part should be rewritten in a more appropriate manner avoiding the mention of previously published literature in this part.

      >> We have revised this by avoiding the mentioning of previously published literature.

1.2.            Minor comments:

5-      Abstract: Page 1, lines 19–20: “We grew… litter”: Kindly avoid the first voice form of the sentence and adopt the impersonal form instead.

>> We have revised this as suggested.

6-      Abstract: Page 1, lines 20–22: “Elevated… treatment”: Kindly mention some percentages of these increases.

>> We have revised this as suggested.

7-      1. Introduction: Page 1, lines 42–44: “However… acquisition”: The sentence is cumbersome; accordingly, kindly reformulate in order to make it clearer and more aiming.

>> We have revised this as suggested by rephrasing trying best to make it clearer.

8-      1. Introduction: Page 2, lines 57–58: “Hence… mushrooms”: The sources of this statement are very old; accordingly, kindly replace them by the following one as a new and reliable source: https://doi.org/10.3390/horticulturae8040316.

      >> We have chiefly revised this as suggested by adding the reference provided by reviewer#2.

9-      1. Introduction: Page 2, lines 64–71: “To overcome… biomass”: Kindly avoid the first voice form of these sentences and adopt the impersonal form instead.

>> We have revised this as suggested by avoiding first person voice.

10-  2. Materials and methods, 2.2. Assessment of mycelial growth and reproductive success: Pages 2–3, lines 92–97: “To gain… plate”: Same recommendation as in the previous comment.

>> We have revised this as suggested by avoiding first person voice.

11-  2. Materials and methods, 2.2. Assessment of mycelial growth and reproductive success: Page 3, line 99: Kindly mention the full specification of the used balance.

      >> We have revised this as suggested.

12-  2. Materials and methods, 2.2. Assessment of mycelial growth and reproductive success: Page 3, lines 101–109: “As indicators… 40 ℃”: Kindly avoid the first voice form of these sentences and adopt the impersonal form instead.

>> We have revised this as suggested by avoiding first person voice.

13-  2. Materials and methods, 2.3. Statistical analysis: Page 3, line 111: Kindly adjust this paragraph’s title as “2.3. Statistical analysis”.

>> We have revised this as suggested.

14-  2. Materials and methods, 2.3. Statistical analysis: Page 3, lines 112–123: Kindly avoid the first voice form of this paragraph’s sentences and adopt the impersonal form instead.

>> We have revised this as suggested by avoiding first person voice.

15-  3. Results: Page 3, lines 125–127: “We found… control”: Same recommendation as in the previous comment.

>> We have revised this as suggested by avoiding first person voice.

16-  3. Results: Page 3, lines 127–129: “There was… cultures”: Kindly remove this sentence as analysing non-significant findings is not very appropriate scientifically.

      >> We have revised this as suggested.

17-  3. Results: Page 3, line 132: Kindly add “respectively” after “8.1 days”.

>> We have rephrased in a way that lifts the necessity of adding “respectively”.

18-  3. Results: Page 3, lines 133–134: “The number… supplementation”: Kindly remove this sentence as analysing non-significant findings is not very appropriate scientifically.

>> We have revised this as suggested.

19-  3. Results: Page 3, lines 136–138: “We found… control”: Kindly avoid the first voice form of the sentence and adopt the impersonal form instead.

>> We have revised this as suggested by avoiding first person voice.

20-  3. Results: Page 3, lines 138–139: “but no… treatments”: Kindly remove this sentence as analysing non-significant findings is not very appropriate scientifically.

>> We have revised this as suggested.

Please note that we followed your request. Still, we think that “negative” results are also worth mentioning and are not less scientific than “positive” results. Also, the “negative” results are displayed in Fig. 1 and thus we think it is ok not to reiterate them in text-form and thus we deleted it from the results section in text form.

21-  4. Discussion: Pages 4–5, lines 152–155: “Here… cycle”: Kindly avoid the first voice of the sentence and adopt the impersonal form instead.

>> We have revised this as suggested by avoiding first person voice.

22-  4. Discussion, 4.1. Effect of adenosine addition on mycelium and primordia formation: Page 5, lines 157–160: “Previous… medium”: The sentence is cumbersome; accordingly, kindly reformulate in order to make it clearer and more aiming.

>> We have revised this as suggested by rephrasing.

23-  4. Discussion, 4.1. Effect of adenosine addition on mycelium and primordia formation: Page 5, lines 160–162: “In overall… medium”: Kindly avoid the first voice form of the sentence and adopt the impersonal form instead.

>> We have revised this as suggested by avoiding first person voice.

24-  4. Discussion, 4.1. Effect of adenosine addition on mycelium and primordia formation: Page 5, lines 164–167: “Similar… (see Fig. 1A)”: Same recommendation as in the previous comment.

      >> We have revised this as suggested by avoiding first person voice.

25-  4. Discussion, 4.1. Effect of adenosine addition on mycelium and primordia formation: Page 5, lines 170–171: Kindly adjust as follow: “in the present case”.

>> We have revised this as suggested.

26-  4. Discussion, 4.1. Effect of adenosine addition on mycelium and primordia formation: Page 5, lines 172–175: “Other… (see Fig. 1B)”: Kindly avoid the first voice form of the sentence and adopt the impersonal form instead.

>> We have revised this as suggested avoiding first person voice.

27-  4. Discussion, 4.1. Effect of adenosine addition on mycelium and primordia formation: Page 5, line 178: Kindly adjust as follow: “appeared”.

      >> We have revised this as suggested.

28-  4. Discussion, 4.1. Effect of adenosine addition on mycelium and primordia formation: Page 5, line 189: Kindly add a comma “,” after “Earlier”. 

      >> We have revised this by replacing “Earlier” with “Accelerated” to make it clearer.

29-  4. Discussion, 4.1. Effect of adenosine addition on mycelium and primordia formation: Page 5, line 193: Kindly adjust as follow: “in comparison with”.     

      >> We have revised this as suggested.

30-  4. Discussion, 4.2. Effect of adenosine addition on reproductive biomass in fruit bodies: Page 5, lines 195–197: “Compared… (see Fig. 1E)”: Kindly avoid the first voice form of the sentence and adopt the impersonal form instead.

>> We have revised this as suggested avoiding first person voice.

31-  4. Discussion, 4.2. Effect of adenosine addition on reproductive biomass in fruit bodies: Page 5, lines 199–202: “It is… (see Fig. 1C)”: The sentence is cumbersome; accordingly, kindly reformulate in order to make it clearer and more aiming.

>> We have revised this as suggested.

32-  4. Discussion, 4.2. Effect of adenosine addition on reproductive biomass in fruit bodies: Page 5, line 203: Kindly adjust as follow: “results of” and “These authors”.

>> We have revised this as suggested.

33-  4. Discussion, 4.3. Study limitations and future perspective: Page 6, line 210: Kindly adjust as follow: “The present initial”.

>> We have revised this as suggested.

34-  4. Discussion, 4.3. Study limitations and future perspective: Page 6, line 211: Kindly replace “contains” by “outcastes”.

>> We have replaced “also contains” by “it also has”.

35-  4. Discussion, 4.3. Study limitations and future perspective: Page 6, lines 217–223: “We here… ScMM”: Kindly avoid the first voice form of these sentences and adopt the impersonal form instead.

      >> We have revised this as suggested avoiding first person voice.

36-  4. Discussion, 4.3. Study limitations and future perspective: Page 6, line 225: Kindly write the Latin name in Italic form.

>> We have revised this as suggested.

37-  4. Discussion, 4.3. Study limitations and future perspective: Page 6, line 230: Kindly adjust as follow: “The present experiment”.

>> We have revised this as suggested.

38-  5. Conclusion: Page 6, lines 243–246: “In this… formation”: Kindly avoid the first voice form of these sentences and adopt the impersonal form instead.

>> We have revised this as suggested avoiding first person voice.

39-  5. Conclusion: Page 6, line 246: Kindly adjust as follow: “The present positive”.

>> We have chiefly revised this as suggested adjusting to “This positive”.

40-  5. Conclusion: Page 6, lines 249–252: “Anthropogenic… webs”: Kindly remove this sentence as mention information related to published literature in the Conclusion part is inappropriate.

>> We have revised this by removing the indicated sentence.

Round 2

Reviewer 2 Report

Comments to the Author:

Title: Organic nitrogen supplementation increases vegetative and reproductive biomass in a versatile white rot fungus

Overview and general recommendation:

Authors made significant improvements to their manuscript and are well thanked for that. I have no more comments for the authors.

Minor comments:

1-      2. Materials and methods, 2.2. Assessment of mycelial growth and reproductive success: Page 3, lines 109–111: “We cut… 40 ℃”: Kindly avoid the first voice form of the sentence and adopt the impersonal form instead.